# Temporal and Spatial Characteristics of Meteorological Elements in the Vertical Direction at Airports and Hourly Airport Visibility Prediction by Artificial Intelligence Methods

**Jin Ding** [1], **Guoping Zhang** [1], **Jing Yang** [1,*], **Shudong Wang** [1,*], **Bing Xue** [1], **Xiangyu Du** [2], **Ye Tian** [3], **Kuoyin Wang** [1], **Ruijiao Jiang** [1] and **Jinbing Gao** [1]

1 Public Meteorological Service Center, China Meteorological Administration, Beijing 100081, China
2 Institute of Information Engineering, Chinese Academy of Sciences, Beijing 100093, China
3 School of Science, Beijing University of Posts and Telecommunications, Beijing 100876, China
* Correspondence: yangjing@cma.gov.cn (J.Y.); stonemon@foxmail.com (S.W.)

**Abstract:** Based on second-level L-band sounding data, the vertical distribution and variation of meteorological elements at airports in 2010–2020 are investigated. At the same time, the relationships between airport visibility and meteorological elements at different potential heights are also investigated. Then, based on hourly measurements of 26 meteorological elements in 2018–2020, the hourly visibility of airports is predicted by 9 artificial intelligence algorithm models. The analyses show: (1) For the vertical changes in four meteorological elements of the airports, the negative vertical trends of temperature and relative humidity increase clearly from northwestern to southeastern China. The significant negative trend of air pressure in the vertical direction in the eastern China is greater. (2) Within about 2000 geopotential metres (gpm) from the ground, the visibility has a strong correlation with the air pressure, and most of them are negative. Within 400 gpm from the ground, airport visibility is negatively correlated with the relative humidity. At 8:00 a.m., airport visibility is positively correlated with the wind speed within 2000 gpm from the ground at most airports, while at 20:00 p.m., the positive correlation mainly appears within 400 gpm from the ground. (3) The passive aggressive regression-(PAR) and isotonic regression-(IST) based models have the worst effect on airport visibility prediction. The dispersion degree of the visibility simulation results obtained by Huber regression-(HBR) and random sample consensus regression-(RANSAC) based models is relatively consistent with the observations.

**Keywords:** airport visibility; prediction; artificial intelligence; L-band sounding data

## 1. Introduction

The observation and analysis of high-altitude meteorological data are of great significance in meteorology. The horizontal movement of the atmosphere characterized by these data, accompanied by the exchange of heat and water vapor, can reveal the specific process of the occurrence, development and evolution of a certain weather situation, so it plays an important role in forecasting. In China, the L-band electronic radiosonde system, which can provide second-order radiosonde positioning data with high vertical resolution, was fully used for exploration in 2011. The detection accuracy of air pressure, temperature and humidity improved greatly after the L-band radar detection system was put into use in the meteorological industry. Based on L-band sounding data, scholars have carried out applied research in many aspects. For severe convective weather, Qing et al. [1] established a prediction model for short-term rainstorm prediction in the Sichuan Basin based on the V-3θ graph made by L-band sounding data. Huang et al. [2] studied the variation characteristics of atmospheric physical quantities based on L-band sounding data and found that there was downdraft between 850 hPa and 500 hPa; additionally, the speed of balloons returned to normal after 500 hPa. By analyzing the humidity observation conditions of the three

main L-band radiosondes used in China and their deviation distribution characteristics, and referring to the corresponding deviation correction tests on the correction methods of radiosonde humidity at home and abroad, Hao et al. [3] developed a correction method suitable for China's L-band radiosondes and applied it in the model assimilation analysis to improve the assimilation and prediction results. According to the characteristics of L-band second-level data, Tian and Zhang [4] made a preliminary analysis on the application prospect in artificial precipitation operation, in order to improve its application rate in artificial precipitation operation. According to the characteristics of L-band sounding data, their application in airport visibility will make a great contribution to the research of airport visibility.

Airport low-visibility events will affect the take-off and landing of aircraft, personnel safety and major economic losses, which makes it urgent to conduct in-depth research on airport visibility. Most studies of airport visibility are mainly to predict the airport visibility through other meteorological elements or air pollution factors. Debashree et al. [5] selected $NO_2$, wind speed, relative humidity, CO, and temperature as parameters and used an artificial neural network (ANN) model to forecast the spatial visibility during fog over the Kolkata airport. Based on air temperature, relative humidity, visibility, wind speed, wind direction, precipitation, soil temperature and soil moisture, a new model for nowcasting visibility in the coastal desert area of Dubai [6]. Based on observations of wind speed and other four meteorological elements from October to March in the last decade, Zhu et al. [7] forecast hourly visibility at Urumqi Airport. Kneringer et al. [8] used an ordered logistic regression model to forecast cold season visibility at Vienna International Airport. Based on meteorological elements and air pollution indicators, Deng et al. [9] used the long-term short-term memory network (LSTM) model and introduced a weighted loss function to forecast low visibility for airports, and the results showed that under appropriate super parameters, the RMSE of visibility decreased by 37% under 1600 m and 21% under 800 m. Won et al. [10] investigated the visibility of Incheon International Airport in South Korea during 2015–2017 and its relationship with PM2.5 and PM10, and then established a truncated regression model to quantitatively describe the change in visibility, which showed that the visibility decline was mainly determined by the interaction between PM2.5 and meteorological factors (such as fog, haze, high temperature, low relative humidity and weak wind speed). Sara et al. [11] discussed an hourly short-term prediction of low-visibility events at Spain Villanubla Airport with PM10, PM2.5, temperature, precipitation, pressure, relative humidity, wind speed and wind direction by Markov chain models and machine learning techniques. Wu et al. [12] used the atmospheric state information of the airport ground station to build a plateau airports visibility prediction model to predict the hourly visibility of the plateau airports in the next 1–6 h by the LSTM. Liu et al. [13] used data-driven depth learning method and multiple nonlinear regression analysis method to analyze the relationship between the runway visual range (RVR) and ground meteorological elements for different reasons and input the pseudo color images converted by the original images into the depth model integrated by two popular convolutional neural network (CNN) models: VGG-16 and Xception for analysis.

According to previous studies, due to the limitation of meteorological data, less kinds of meteorological elements can be used, and only the most common individual meteorological elements are used to predict visibility. At the same time, almost no scholars try to explore the relationship between airport visibility and L-band sounding data. In view of the above problems, on the one hand, this study attempts to analyze the temporal and spatial characteristics of L-band sounding data and explore their relationship with airport visibility. On the other hand, this study will use a variety of meteorological elements to further explore the short-term approach forecast of airport visibility combined with artificial intelligence methods.

## 2. Methodology

### 2.1. Data

Forty-seven international airports in China were selected as research objects to study the relationship between L-band sounding data and visibility and explore hourly visibility prediction. Among them, 22 airports own the second level L-band sounding data. Figure 1 and Table 1 show the locations of the airports and whether they have L-band sounding data.

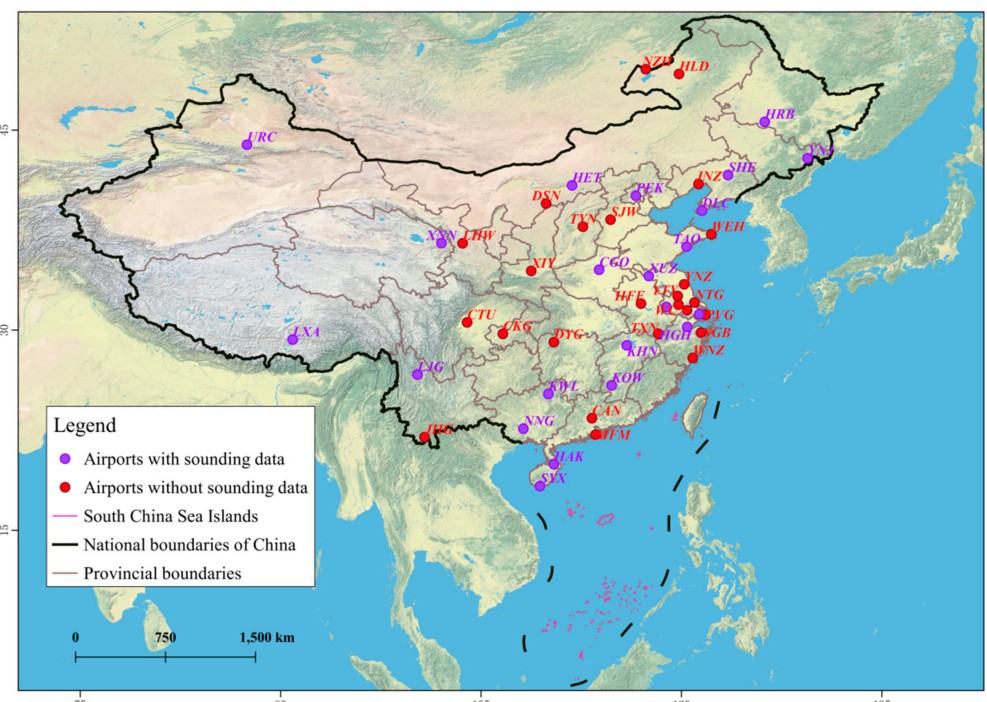

**Figure 1.** The locations of the selected 47 international airports in China. Purple points and red points indicate airports with L-band sounding data and without L-band sounding data, respectively.

L-band high-altitude meteorological detection is coordinated by secondary wind radar and electronic radiosonde. It mainly detects high-altitude wind direction, wind speed, air temperature, air pressure and other meteorological elements. The sounding data are detected twice a day at 8:00 a.m. and 20:00 p.m. Beijing Time (BJT). L-band can provide sounding data with high vertical resolution (the sampling period of electronic radiosonde can reach 1.2 s). Observed second level L-band sounding data is obtained from the website of Climate Data Center of China Meteorological Administration (CMA; https://data.cma.cn; last accessed on 17 August 2022). The data has undergone strict quality control, such as allowable value range check, rigid value check, vertical consistency check, climate limit value check, filter check, monotonicity check, consistency check between elements, etc. Since the quality control scheme of second level data is not the focus of this study, it will not be given unnecessary details here. Based on the second level L-band sounding data during 2010–2020, the mean of the data per minute is calculated to convert the second level data into minute level data. As the different detection duration for each detection, the first 60 min of each sounding data of the airport are selected for the unified analysis.

Hourly meteorological elements are obtained from the CMA (https://data.cma.cn, accessed on 21 September 2022). From more than 92,000 national and regional weather stations in China, the visibility average after data quality control of the five stations closest to the airport is selected as the actual visibility on the airport. For data quality control, missing, mutation point and abnormal hourly data are detect first, and then the abnormal values are deleted, and the missing values are interpolated. Considering the required high time resolution and long time series, the hourly 26 meteorological elements in 2018–2020

are selected to form the datasets required for visibility forecast. The elements contained in second level sounding data and hourly meteorological elements are shown in Table 2.

**Table 1.** Information about the selected international airports.

| Airports with L-Band Sounding Data | | | | Airports without L-Band Sounding Data | | | |
|---|---|---|---|---|---|---|---|
| **ID** | **Code** | **Name** | **City** | **ID** | **Code** | **Name** | **City** |
| 1 | CGO | Xinzheng | Zhengzhou | 1 | CAN | Baiyun | Guangzhou |
| 2 | DLC | Zhoushuizi | Dalian | 2 | CKG | Jiangbei | Chongqing |
| 3 | HAK | Meilan | Haikou | 3 | CTU | Shuangliu | Chengdu |
| 4 | HET | Baita | Hohhot | 4 | CZX | Benniu | Changzhou |
| 5 | HGH | Xiaoshan | Hangzhou | 5 | DSN | Yijinhuoluo | Erdos |
| 6 | HRB | Taiping | Harbin | 6 | DYG | Hehua | Zhangjiajie |
| 7 | KHH | Gaoxiong | Gaoxiong | 7 | HFE | Xinqiao | Hefei |
| 8 | KOW | Huangjin | Ganzhou | 8 | HLD | Dongshan | Hailar |
| 9 | KWL | Liangjiang | Guilin | 9 | JHG | Gasa | Xishuangbanna |
| 10 | LJG | Sanyi | Lijiang | 10 | JNZ | Jinzhou | Jinzhou |
| 11 | LXA | Gongga | Lhasa | 11 | LHW | Zhongchuan | Lanzhou |
| 12 | NKG | Lukou | Nanjing | 12 | MFM | Aomen | Macao |
| 13 | NNG | Wuwei | Nanning | 13 | NGB | Lishe | Ningbo |
| 14 | PEK | Shoudu | Beijing | 14 | NTG | Xingdong | Nantong |
| 15 | SHA | Honqiao | Shanghai | 15 | NZH | Xijiao | Manzhouli |
| 16 | SHE | Taoxian | Shenyang | 16 | PVG | Pudong | Shanghai |
| 17 | SYX | Fenghuang | Sanya | 17 | SJW | Zhengding | Shijiazhuang |
| 18 | TAO | Liuting | Qingdao | 18 | TXN | Tunxi | Huangshan |
| 19 | URC | Diwobu | Urumqi | 19 | TYN | Wusu | Taiyuan |
| 20 | XMN | Gaoqi | Xiamen | 20 | WEH | Dashuipo | Weihai |
| 21 | XUZ | Guanyin | Xuzhou | 21 | WNZ | Longwan | Wenzhou |
| 22 | YNJ | Chaoyangchuan | Yanji | 22 | WUH | Tianhe | Wuhan |
| | | | | 23 | XIY | Xianyang | Xian |
| | | | | 24 | YNZ | Nanyang | Yancheng |
| | | | | 25 | YTY | Taizhou | Yangzhou |

**Table 2.** The details of the minute sounding data and hourly meteorological data.

| Elements of the Hourly Meteorological Data | | Elements of the Minute Sounding Data |
|---|---|---|
| Air temperature (°C) | Pressure (hPa) | Air temperature (°C) |
| Minimum temperature (°C) | Sea level pressure (hPa) | Pressure (hPa) |
| Maximum temperature (°C) | Vapor pressure (hPa) | Relative humidity (%) |
| Dew point temperature (°C) | 2-min average wind speed (m/s) | Wind speed (m/s) |
| Relative humidity (%) | 10-min average wind speed (m/s) | Wind direction (degree) |
| Precipitation in the past hour (mm) | Maximum wind speed (m/s) | Geopotential hight (gpm) |
| Precipitation in the past 6 h (mm) | Wind direction of maximum wind speed (degree) | |
| Precipitation in the past 12 h (mm) | Extreme instantaneous wind speed (m/s) | |
| Ground surface temperature (°C) | Direction with extreme wind speed (degree) | |
| Ground temperature at 5 cm depth (°C) | Maximum instantaneous wind speed in the past 6 h (m/s) | |
| Ground temperature at 10 cm depth (°C) | Direction of Maximum instantaneous wind speed in the past 6 h (degree) | |
| Ground temperature at 15 cm depth (°C) | Maximum instantaneous wind speed in the past 12 h (m/s) | |
| Ground temperature at 20 cm depth (°C) | Direction of Maximum instantaneous wind speed in the past 12 h (degree) | |

### 2.2. Methods

### 2.2.1. Trend Test

The Mann–Kendall (MK) test, whose advantage is that it can test linear or nonlinear trend, is a rank-based nonparametric test [14,15]. In the test, the null hypothesis ($H_0$) and the alternative hypothesis ($H_1$) are equal on whether there is a time-series trend in the observed data. In the MK test, the statistical value S and the standardized test statistical value Z are calculated as follows:

$$S = \sum_{i=1}^{n-1} \sum_{j=i+1}^{n} \text{sgn}(X_j - X_i) \tag{1}$$

$$\text{sng}(X_j - X_i) = \begin{cases} +1, & \text{if } (X_j - X_i) > 0 \\ 0, & \text{if } (X_j - X_i) = 0 \\ -1, & \text{if } (X_j - X_i) < 0 \end{cases} \tag{2}$$

$$\text{Var}(S) = \frac{1}{18}\left[n(n-1)(2n+5) - \sum_{p=1}^{q} t_p(t_p - 1)(2t_p + 5)\right] \tag{3}$$

$$Z = \begin{cases} \frac{S-1}{\sqrt{\text{Var}(S)}} & \text{if } S > 0 \\ 0 & \text{if } S = 0 \\ \frac{S+1}{\sqrt{\text{Var}(S)}} & \text{if } S < 0 \end{cases} \tag{4}$$

where $X_i$ and $X_j$ correspond to i and j in the time series; n is the length of the time-series data; $t_p$ is the binding value corresponding to the number p; Z is the change trend of time series data. If $Z > 0$, it means that the time series data shows an increasing trend with the passage of time; Otherwise, it means that the time series data show a decreasing trend with the passage of time. When $|Z| > Z_{(1-\alpha/2)}$, the null hypothesis is rejected and it is considered that there is a significant trend in the time series data. $Z_{(1-\alpha/2)}$ can be calculated from the standard normal distribution table. When the significance level $\alpha = 5\%$, the corresponding value of $Z_{(1-\alpha/2)}$ is 1.96.

The nonparametric Sen's slope [16] is used to quantify the monotonic trends [17]. The advantage of this method is that it limits the influence of external interference on the trend of time series data.

$$\beta = \text{Median}\left(\frac{X_j - X_i}{j - i}\right) \tag{5}$$

where for any $i < j$, $X_i$ and $X_j$ are the values corresponding to i and j in the time series data; $\beta$ is the calculated value of the trend size of time series data.

### 2.2.2. Artificial Intelligence Algorithms

### Isotonic Regression (IST)

IST is a regression model that makes nonparametric estimation on given data in monotonic function space [18]. A finite set of real numbers $Y = y_1, y_2 \cdots y_n$ and $X = x_1, x_2 \cdots x_n$ are given to train a model to minimize the following equation:

$$f(x) = \sum_{i=1}^{n} \omega_i(y_i - x_i)^2 \tag{6}$$

where $x_i \leq x_2 \cdots \leq x_n$, $\omega_i$ are positive weights. In this study, the pool adjacent violators algorithm (PAVA) is used to simplify IST, making it an unweighted linear sequence IST. Roughly speaking, the process of PAVA algorithm is described as follows: Starting from $y_1$ on the left, moving $y_1$ to the right until the first violation are encountered, namely $y_i < y_i + 1$. Then we replace these y before the violation with $y^2$ to meet the monotonicity. Then this process will continue until reaching $y_n$.

Bayesian Ridge Regression (BRR)

BRR is proposed based on an improved Bayesian method and least squares [19,20]. This method adds L2 regularization to the linear Bayesian regression model, combines the prior information of relevant parameters to form a prior distribution and gives the estimated value. The function of Bayesian linear regression is shown in Equation (7), which aims to find the parameter vector distribution that minimizes the loss function Equation (8):

$$y(x, \omega) = \sum_{j=0}^{n} \omega_j \varphi_j(x) = \omega^T \varphi(x) \tag{7}$$

$$J(\omega) = \sum_{i=1}^{m} \{y(x_i, \omega) - t_i\}^2 \tag{8}$$

where n is the dimension of sample space; m is the sample capacity; $\omega$ is a parameter vector; $\varphi(x)$ is a nonlinear function of the input vector x, where $\varphi_0(x) = 1$, $t_i$ is the observed value, $t_i = y(x_i, \omega) + \varepsilon$, $\varepsilon$ is noise, assuming $\varepsilon$ and $\omega$ obey Gaussian distribution $N(0, \sigma_1^2)$ and $N(0, \sigma_2^2)$ respectively, t obeys the Gaussian distribution of the mean $y(x, \omega)$. The class conditional probability density function of t is Equation (9), The prior probability of $\omega$ is Equation (10):

$$p(t|\omega) = \frac{1}{2\pi\sigma_1{}^2} \exp\left(-\frac{1}{2\sigma_1^2} \sum_{i=1}^{m} \{y(x_i, \omega) - t_i\}^2\right) \tag{9}$$

$$p(\omega) = \frac{1}{2\pi\sigma_2{}^2} \exp\left(-\frac{1}{2\sigma_2^2} \omega^T \omega\right) \tag{10}$$

According to Bayesian rules:

$$p(\omega|t) = \frac{p(\omega)p(t|\omega)}{p(t)} \tag{11}$$

$$\ln(p(\omega|t)) = -\frac{1}{2\sigma_1^2} \sum_{i=1}^{m} \{y(x_i, \omega) - t_i\}^2 - \frac{1}{2\sigma_2^2} \omega^T \omega + c \tag{12}$$

where $p(\omega|t)$ is a posteriori probability; $p(t)$ is a constant independent of $\omega$; C is constant. BRR automatically introduces the regular term in the estimation process, and finally obtains the posterior distribution of parameters, which avoids over fitting in maximum likelihood estimation and obtains more accurate parameter estimation.

Least Absolute Shrinkage and Selection Operator (LASSO)

LASSO is a kind of compression estimation [21]. It constructs a penalty function to obtain a more refined model, which makes it compress some regression coefficients; that is, the sum of the absolute values of the forced coefficients is less than a fixed value. At the same time, some regression coefficients are set to zero. Therefore, it retains the advantage of subset contraction and is a biased estimation for processing complex collinear data.

In ordinary linear regression, the objective of optimization is to minimize the sum of squares of residuals. However, when the number of independent variables is large, or there is a multicollinearity problem between variables, it is easy to lead to cumbersome solution of the generalized inverse matrix, complex model, and finally over fitting. The regularization method is mainly used to solve the above problems. On the basis of the loss function, a penalty function related to the coefficient is added to constrain the complexity of the model, so as to select the model with less empirical risk and model complexity at the same time. The optimized objective function is:

$$J(\beta) = \sum_{i-1}^{N} \left(y_i - \alpha_i - \sum_{j-1}^{P} x_{ij}\beta_j\right)^2 + \lambda f(\beta) \tag{13}$$

where $y_i$ represents the dependent variable, $\alpha_i$ represents the offset term, $x_{ij}$ represents the independent variable, $\beta_j$ is a coefficient, N is the number of data groups, P is the number of independent variables, $\lambda \geq 0$ represents the coefficient to balance the regularization term and sum of squared residual and $f(\beta)$ is the penalty function related to coefficient matrix $\beta$.

LASSO sets penalty function $f(\beta)$ as the L1 norm for coefficient matrix $\beta$ and ignores the influence of the bias term $\alpha_i$ on the final result. In order to increase the proportion of regularization term in the optimization objective function and strengthen the constraint of coefficients, the sum of squared residual in Equation (13) is averaged and added to the regularization term; then, the optimization objective function of LASSO algorithm can be shown as:

$$J(\beta) = \frac{1}{N} \sum_{i-1}^{N} \left( y_i - \sum_{j-1}^{P} x_{ij}\beta_j \right)^2 + \lambda ||\beta||_1 \tag{14}$$

L1 norm is to sum the absolute values of each element of the corresponding vector. Therefore, the optimization objective function of LASSO algorithm is discontinuous and differentiable. Generally, the gradient descent methods are used to solve it.

Passive Aggressive Regression (PAR)

PAR embodies an idea of online learning [22], which can continuously integrate new samples to adjust the classification model and enhance the classification ability of the model. PAR optimizes its own model according to the following judgment criteria: (A) The model will not be adjusted when the sample classification is correct and the model predicts the possibility accurately (more than one degree); (B) when the sample classification is correct but the model's prediction of probability is biased (not very accurate), the model makes a slight adjustment; (C) the model makes a large adjustment when the sample classification is wrong. In this process, different samples with different weights can gradually approach a suitable parameter to represent the relationship between different features.

Random Sample Consensus Regression (RANSAC)

RANSAC adopts the iterative method to estimate the mathematical model from a group of discrete observation data and improves the probability by increasing the number of iterations [23]. RANSAC uses the optimization method to select the parameters of the model, that is, first build the cost function, and then determine the model parameters by maximizing the cost function. The specific model is as follows:

$$\hat{\theta} = \arg\max \left\{ \sum^{\varphi \in \vartheta} J[\rho(\varphi, \theta)] \right\} \tag{15}$$

where $\theta$ is the model parameter set, $\varphi$ is the feature point set, $\rho$ Is the error function, J represents the cost function. In the linear detection problem, j is the number of feature points in the uniform set.

RANSAC randomly selects subsets of observation data. The number of elements in the subsets is the minimum number of data required to determine the model n. the selected subsets are assumed to be local points, and the algorithm is completed according to the following steps: (A) A model is determined by the assumed local point, and the relevant unknown parameters are determined by the local point and the model. (B) Use the model obtained in Step A to test other data. If the linear distance from a point to the model is within the threshold range, it is judged that the point is an internal point. (C) The estimated model is considered to be reasonable enough, if enough points are judged as local points. (D) All local points are used to re estimate the model, and this model is used as the final model of the data. The whole process is an iteration, and the best fitting model is selected by a fixed number of iterations.

Huber Regression (HBR)

Huber loss function is that L2 loss function is used when the prediction deviation is less than the robustness parameter; When the prediction deviation is greater than the robustness parameter, L1 loss function is used. Huber regression does not ignore the existence of outliers and uses linear loss for outliers [24]. Compared with L2 loss, Huber regression reduces the weight of outliers and ultimately reduces the impacts of outliers on regression. Subsequently, Huber and Peter [25] proved the large sample asymptotic property of the parameter estimator of Huber regression and made it widely used in many fields. The optimization objective function of Huber regression is as follows:

$$\min_{\omega,\sigma} \sum_{i=1}^{n} (\sigma + H_\varepsilon (\frac{X_i \omega - y_i}{\sigma})\sigma) + \alpha ||\omega||_2^2 \tag{16}$$

where

$$H_\varepsilon(z) = \begin{cases} z^2, & if |z| < \epsilon \\ 2\epsilon|z| - \epsilon^2 & otherwise \end{cases} \tag{17}$$

Elastic-Net Regression (ENR)

When multiple features are correlated, LASSO may choose only one at random, while Ridge regression [26] will choose all features. At this time, it is easy to think that if these two regularization methods are combined, the disadvantages of the two methods may be combined. This regularized algorithm is called ENR. ENR is a linear regression model using L1 and L2 priors as regularization matrices [27]. ENR automatically selects features while maintaining the shrinkage coefficient. It can select group related features, or select more features than the number of samples to saturate the features. It is suitable for models with multiple features related to each other. ENR is defined as follows:

$$\operatorname{argmin}[\sum_{i=1}^{N}(y_i - \hat{y}_i)^2 + \gamma\alpha \sum_{j=1}^{k} ||\beta_j||_1 + \frac{\gamma(1-\alpha)}{2} \sum_{i=1}^{N} ||\beta_j||_2^2] \tag{18}$$

where $\gamma$ is a complex parameter that controls the degree of compression (0 means no penalty, $\infty$ means full penalty), $\alpha (0 \leq \alpha \leq 1)$ is the mixed parameter of elastic network, $||\beta_j||_2^2$ is the ridge regression term, $||\beta_j||_1$ is the lasso item.

Automatic Relevance Determination Regression (ARD)

ARD is a sparse model obtained in data training based on Bayesian principle applied to regression problems [28]. Let the sample data set used for training be $\{x_n, t_n \mid n = 1, 2, \cdots N\}$, $x_n$ represents the sample input value used for training, $t_n$ represents the output of independent distribution, and establish the functional relationship between $x_n$ and $t_n$:

$$t_n = y(x_n; \omega) + \xi_n \tag{19}$$

where $\xi_n$ satisfying the additional Gaussian noise for $\xi_n \sim N(0, \sigma^2)$, $\sigma^2$ is the quantity to be solved, and it can be inferred that Equation (9) satisfies the Gaussian distribution:

$$p(t_n|x) = N(t_n \mid y(x_n), \sigma^2) \tag{20}$$

Since $t_n$ does not interfere with each other and is independent of each other, the likelihood function of the training sample set can be expressed as follows:

$$p(t|\omega, \sigma^2) = 2\pi\sigma^{2-\frac{N}{2}} \exp\{-\frac{1}{2\sigma^2} ||t - \theta\omega||^2\} \tag{21}$$

where $\omega = [\omega_0, \omega_1, \cdots \omega_N]^T$, $\theta$ is $n \times (n+1)$ matrix, $\omega_i$ satisfies the Gaussian distribution of the prior distribution that the mean is 0 and the variance is $\alpha_i^{-1}$. Hyperparameter

$\alpha = [\alpha_0, \alpha_1, \cdots, \alpha_N]^T$, $\alpha_i$ corresponds to one $\omega_i$. The prior distribution can be obtained from the training sample data set. The likelihood function of the training sample set is determined by Equation (21). According to the Bayesian principle, the weight value $\omega_i$, the expression of the posterior distribution, can be obtained as follows:

$$p(t|\omega, \alpha, \sigma^2) = \begin{cases} \frac{P(t|\omega,\sigma^2)P(\omega,\alpha)}{P(t|\alpha,\sigma^2)} \\ (2\pi)^{-(N+1)/2}|\textstyle\sum|^{-1/2}\exp\left\{-\frac{1}{2}\omega - m^T\textstyle\sum^{-1}(\omega - m)\right\} \end{cases} \tag{22}$$

where $m = \sigma^2\sum\theta^T t$, $\sum = \left(\sigma^{-2}\theta^T\theta + A\right)^{-1}$, $A = \text{diag}(\alpha_0, \alpha_1, \cdots, \alpha_N)$. The maximum likelihood function is:

$$p(t|\alpha, \sigma^2) = \begin{cases} \int P(t|\omega, \sigma^2)P(\omega|\alpha)d\omega \\ (2\pi)^{-(N+1)/2}|C|^{-1/2}\exp\left\{\frac{1}{2}t^T C^{-1}t\right\} \end{cases} \tag{23}$$

where $C = \sigma^2 I + \theta A^{-1}\theta^T$, C is the covariance. After calculating the partial derivatives of $\alpha$ and $\sigma^2$ and making their values equal to 0, the following two formulas can be obtained.

$$\alpha_i^{new} = r_i / \mu_i^2 \tag{24}$$

$$\left(\sigma^2\right)^{new} = \frac{||t - \theta\mu||^2}{N - \sum_i^N r_i} \tag{25}$$

where: $\mu_i$ represents the ith posterior average weight, and $r_i$ is the element on the ith main diagonal. In the calculation process, the values of m and $\Sigma$ are updated with the iteration until the convergence condition or the maximum number of iterations are satisfied.

Tweedie Regression (TWD)

Tweedie distribution family is a kind of exponential divergence model, which generally uses $T_{W_P}(\vartheta, \varphi)$, where $\vartheta$ Is the specification parameter, and $\varphi$ is the discrete parameter [29]. A Tweedie distribution family is completely determined by its variance function $V(\mu) = \mu^P$. P is taken from $(-\infty, 0) \cup [1, +\infty)$. It includes several common important distributions as its special cases: p = 0, 1, 2, 3 correspond to normal distribution, Poisson distribution, Gamma distribution and inverse Gaussian distribution respectively. Take Tweedie distribution as the distribution of dependent variable to establish a generalized linear model as follows:

$$y_i \sim T_{W_P}(\vartheta_i, \varphi_i) \tag{26}$$

$$\mu_i = E(y_i) \tag{27}$$

$$g(\mu_i) = x_i'\beta \tag{28}$$

where $x_i = \left(x_{i1}, \cdots, x_q\right)^T$ is a vector composed of q classification variables, T represents transpose, and $\beta$ represents a parameter vector of order q × 1.

Table 3 shows the information of the nine methods and their applicability descriptions. The implementation of the above 9 algorithms all relies on Python programming and its algorithm libraries. Among the nine algorithms, except BRR algorithm [30] applied in the field of visibility prediction, other algorithms are not widely used in airport visibility prediction. This study hopes to try these unused methods.

**Table 3.** The information of 9 artificial intelligence algorithm models and their applicability.

| | Model Names | Applicability |
|---|---|---|
| 1 | Isotonic regression (IST)-based model | IST can find a non decreasing approximation function on the training data while minimizing the mean square error. The advantage of this model is that it does not assume any form of objective function. Isotonic expression has no requirements on the output characteristics of the model and is applicable to the case of large sample size. However, it is easy to over fit when the sample size is small. IST is usually used as an auxiliary method to repair the uneven correction results caused by data sparsity. |
| 2 | Bayesian Ridge Regression (BRR)-based model | BRR is a special case of Bayesian linear regression and belongs to Ridge regression. It has all the characteristics of ridge regression and Bayesian linear regression. |
| 3 | Least absolute shrinkage and selection operator (LASSO)-based model | LASSO is a method that can establish a generalized linear model and filter variables, which is powerful and effective. At the beginning, this statistical model was applied in the field of geophysics, and later it was applied to model building in the medical field. As LASSO has the function of "variable selection", it is often used in traditional low dimensional data in economics. |
| 4 | Passive Aggressive Regression (PAR)-based model | PAR embodies an idea of online learning, which can continuously integrate new samples to adjust the classification model and enhance the classification ability of the model. |
| 5 | Random sample consensus Regression (RANSAC)-based model | RANSAC can estimate the parameters of the mathematical model through iteration from a set of observation data containing outliers. It is an uncertain algorithm—it has a certain probability to get a reasonable result; In order to improve the probability, the number of iterations must be increased. Its advantage is that it can estimate model parameters robustly, but its disadvantage is that only a certain probability can get a credible model, and the probability is proportional to the number of iterations.RANSAC is commonly used in computer vision. |
| 6 | Huber Regression (HBR)-based model | HBR model depends on M-estimate. Compared with the mean, the measurement is less sensitive to outliers. HBR does not ignore outlier, and the linear loss of outlier is adopted, which relatively reduces the weight of outlier and ultimately reduced the impact of outlier on the regression results. And HBR should be faster than RANSAC. |
| 7 | Elastic-Net Regression (ENR)-based model | When multiple features are related, LASSO can only randomly select one of them, while Ridge regression will select all features. ENR can combine the advantages of the two regularization methods, making this method very useful when many features are interrelated. The best thing about ENR is that they can always produce efficient solutions. Since it does not generate cross paths, the resulting solutions are quite good. |
| 8 | Automatic Relevance Determination Regression (ARD)-based model | The maximum likelihood method is used to optimize the parameters, which can infer the relative importance of different inputs from the data. This is an example of ARD. The model is suitable for real-time operation and has been applied to earthquake early warning, earthquake ground motion attenuation estimation and structural health monitoring |
| 9 | Tweedie Regression (TWD)-based model | TWD distribution is a compound distribution of Poisson distribution and gamma distribution. One of the most obvious characteristics of TWD distribution is to generate samples with a value of 0 with a certain probability. This method is often used to analyze semi-continuous data composed of zero and positive continuous data, which widely exists in actuarial science, geosciences and other fields. |

### 2.2.3. The Kurtosis and Skewness Coefficient

Comparing the shape property is is an effective method to test the simulations, as shape property are related to the extreme frequency and amplitude of the data [31]. Kurtosis and skewness are two important measures. Kurtosis is a statistic that studies the steep or smooth distribution of data. Through the kurtosis coefficient, whether the data is steeper or smoother than the normal distribution can be determined. Skewness is a statistic that studies the symmetry of data distribution. Through the skewness coefficient, the degree and direction of asymmetry of data distribution can be determined.

The formula of kurtosis coefficient is as follows:

$$K = \frac{\frac{1}{n}\sum_{i=1}^{n}\left[\left(\frac{X_i - \mu}{\sigma}\right)^4\right]}{\left(\frac{1}{n}\sum_{i=1}^{n}\left[(X_i - \mu)^2\right]\right)^2} \tag{29}$$

where $\mu$ is the mean value and $\sigma$ is the standard deviation. The kurtosis coefficient of the data that completely obey the normal distribution is 3. The larger the kurtosis coefficient, the higher and sharper the probability distribution diagram is, and the smaller the kurtosis coefficient is, the fatter it is.

The formula of skewness coefficient is as follows:

$$S = \frac{E[x^3] - 3\mu\sigma^2 - \mu^3}{\sigma^3} \tag{30}$$

where $\mu$ is the mean value, $\sigma$ is the standard deviation and $E[x^3]$ is the 3-order origin moment of $x$. When S $< 0$, the probability distribution is left biased. When S $= 0$, it means that the data is relatively evenly distributed on both sides of the average value, which is not necessarily an absolute symmetrical distribution. When S $> 0$, the probability distribution diagram is biased to the right.

## 3. Results

### 3.1. Vertical Distribution and Variation of Meteorological Elements at Airports

For the four elements (temperature, air press, relative humidity and wind speed) of L-band sounding data, the vertical distribution of the mean observed twice at 8:00 a.m. and 20:00 p.m. is basically the same in 2010–2020 (Figure 2). Temperature observed at 8:00 a.m. and at 20:00 p.m. ranges from −77.9 °C in the 47th minute at HAK in Haikou to 23.1 °C in the 0th minute at HAK and ranges from −78.3 °C in the 47th minute at HAK to 24.8 °C in the 0th minute at HAK, respectively (Figure 2a,e). The maximum geopotential height that sounding balloon at all stations can reach in the 60th minute is 27,128.4 geopotential metre (gpm) and 27,274.7 gpm at 8:00 a.m. and 20:00 p.m., respectively. In general, within the above two geopotential height ranges, the sounding data at the stations decrease with the increase in geopotential height. However, for most stations, the lowest temperature does not appear in the 60th minute or so but in the 45th- to 55th-minute period. This is related to the detector gradually reaching the stratosphere troposphere junction and even entering the stratosphere. The heat in the troposphere mainly comes from the radiation of the earth's surface, and the temperature of the troposphere decreases with increasing height. The main heat of the stratosphere comes from the ultraviolet rays in the solar radiation absorbed by the original ozone: Therefore, the higher the height, the more solar radiation received, and the temperature increased. In addition to altitude, geographical regions also affect the temperature of vertical detection. For example, the vertical temperature of SYX at Sanya and HAK at Haikou in the tropics is slightly higher than that of other geographical regions.

Air press observed at 8:00 a.m. and at 20:00 p.m. ranges from 19.1 hPa in the 60th minute at LXA in Lhasa to 1016.7 hPa in the 0th minute at SHA in Shanghai and ranges from 18.8 hPa in the 60th minute at LXA to 1016.0 hPa in the 0th minute at SHA, respectively (Figure 2b,f). As the geopotential height increases, the air pressure decreases. In Lhasa, Lijiang and other high-altitude airports, due to the influence of altitude, the 0th minute air pressure is significantly lower than that of other airports. Relative humidity observed at 8:00 a.m. and at 20:00 p.m. ranges from 2.0% in the 60th minute at LJG in Lijiang to 90.9% in the 0th minute at SYX in Sanya and ranges from 2.0% in the 60th minute at LJG to 90.6% in the 0th minute at SYX, respectively (Figure 2c,g). Similar to temperature and air pressure, the relative humidity also changes with the change in geopotential height; that is, the relative humidity decreases with the increase in geopotential height. SYX, HAK and other airports located in the tropical monsoon region have high humidity near the ground due to climate reasons. LXA, located on the Tibetan Plateau, shows a pattern that

the relative humidity increases first and then decreases. Unlike other airports, LXA shows a pattern of relative humidity increasing first and then decreasing, especially at 20:00 p.m. Wang et al. [32] found that the cloud base height at the entrance of the water vapor channel of Yarlung Zangbo Grand Canyon in dry and rainy seasons is mainly 0–1 km and 2–3 km, and more than 40% of the cloud base height is less than 1 km, which is the reason for more precipitation in this region. A large number of humid low clouds may also be the reason why the relative humidity in LXA rises first and then decreases. Wind speed observed at 8:00 a.m. and at 20:00 p.m. ranges from 0.4 m·s$^{-1}$ in the 0th minute at XNN in Caojiabao to 45.5 m·s$^{-1}$ in the 33rd minute at SHA and ranges from 1.5 m·s$^{-1}$ in the 0th minute at KOW in Ganzhou to 45.5 m·s$^{-1}$ in the 33rd minute at SHA, respectively (Figure 2d,h). Unlike temperature, air pressure and relative humidity, wind speed does not have a simple positive or negative correlation with geopotential height, and the stronger wind speed mainly concentrate in the 20th–50th minute period. The wind speed at each airport has little difference near the ground, and the larger wind speed mainly occurs at low altitude airports, such as SHA, NKG in Nanjing and XUZ in Xuzhou. Due to the influence of uneven ground near the ground, the friction slows down the flow of air and the wind is weak. With the increase in geopotential height, the friction force of air becomes small and the wind force gradually increases. Below the tropopause, wind speed usually increases with height and reaches the maximum at the tropopause. While wind speed above the tropopause decreases with the increase in geopotential height.

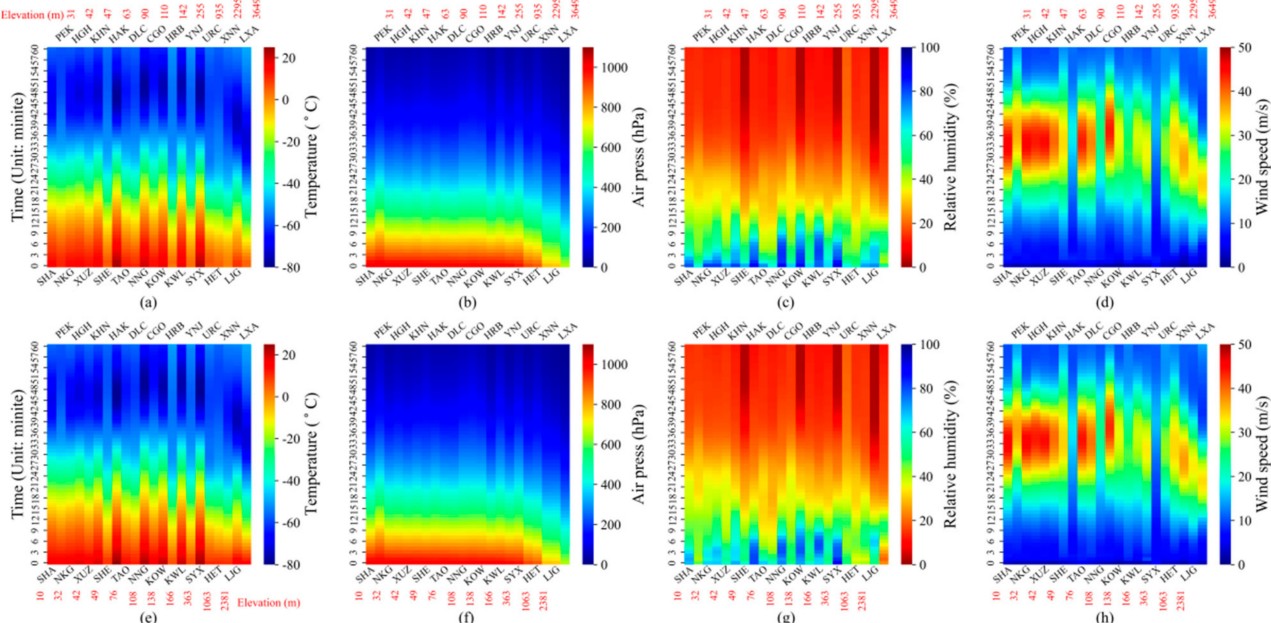

**Figure 2.** Vertical distribution of the mean of temperature (**a**,**e**), air pressure (**b**,**f**), relative humidity (**c**,**g**) and wind speed (**d**,**h**) within 0th−60th minutes of sounding balloon rise during 2010−2020. The upper and lower lines represent data detected at 8:00 a.m. and 20:00 p.m., respectively. The airports are arranged left-to-right by altitude along the *x*-axis.

At 8:00 a.m. and 20:00 p.m., the vertical changes in each meteorological element in 0th–60th minutes are basically the same (Figure 3). The negative vertical trend of temperature has an obvious pattern of increasing from northwestern to southeastern China, and all of them are statistically significant ($p < 0.05$) (Figure 3a,b). The vertical trends of temperature range from −2.04 °C/min at HAK to −0.97 °C/min at HRB and from −2.06 °C/min at HAK to −0.97 °C/min at HRB at 8:00 a.m. and 20:00 p.m., respectively. The vertical trends of air press range from −15.80 hPa/min at PEK to −9.21 hPa/min at LXA and from −15.95 hPa/min at PEK to −9.12 hPa/min at LXA at 8:00 a.m. and 20:00 p.m., respectively (Figure 3d,e). In eastern and western China, the vertical trend range of air

pressure is different, although both are significantly decreased, and the negative trend in the eastern is greater. In the Tibetan Plateau and its surroundings with higher altitude and lower air pressure, the vertical change in air pressure is small. The vertical trends of relative humidity range from −1.30%/min at KOW to −0.48%/min at LXA and from −1.25%/min at KOW to −0.27%/min at LXA at 8:00 a.m. and 20:00 p.m., respectively (Figure 3g,h). With the increase in geopotential height, the relative humidity decreased significantly at all airports. Similar to temperature, the vertical downward trend of relative humidity also shows a decreasing pattern from southeastern to northwestern China in space. The vertical trends of wind speed range from −0.08 m·s$^{-1}$/min at LJG to 0.57 m·s$^{-1}$/min at CGO and from −0.08 m·s$^{-1}$/min at LJG to 0.56 m·s$^{-1}$/min at CGO at 8:00 a.m. and 20:00 p.m., respectively (Figure 3j,k). The regularity of the vertical change in wind speed is not as strong as the above three meteorological elements. On the one hand, the vertical change at all airports is not statistically significant, on the other hand, there is no obvious spatial distribution pattern for wind speed. The wind speed over most airports increases with the increase in geopotential height within 0th–60th minutes, but the wind speed of individual airports decreases with the increase in geopotential height, especially the airports in Hainan Island. Low altitude location and strong sea breeze near the ground may lead to different performances of airports in Hainan. The different performance of URC should also be related to the perennial more sandstorms near the ground in this area. From the relationship between the vertical trends in various meteorological elements and the altitude of the airports, the vertical change in air pressure has a greater relationship with the altitude of the airport, which may also be the reason for its east-west distribution in space (Figure 3c,f,i,l).

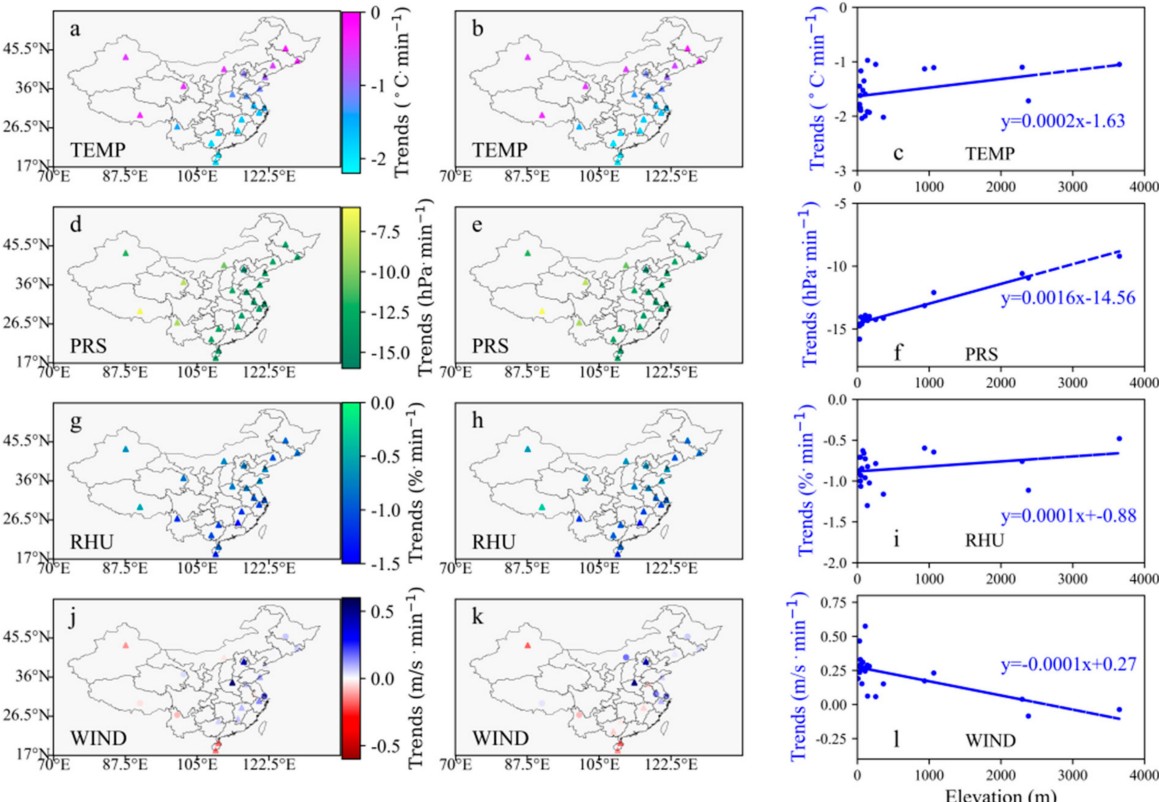

**Figure 3.** Vertical trend of 4 meteorological elements within 0th–60th minutes of sounding balloon rise (**a,b,d,e,g,h,j,k**) and the relationship between the vertical trends and the elevations of the L-band sounding stations (**c,f,i,l**). The left and middle columns represent trends at 8:00 a.m. and 20:00 p.m., respectively. The right column shows the relationship between the trends and elevations at 8:00 a.m. The relationship at 20:00 p.m. is not shown here as it is basically the same as 8:00 a.m. Triangles represent statistically significant trends ($p < 0.05$).

### 3.2. Relationship between Meteorological Elements and Visibility at Different Geopotential Heights

Whether at 8:00 a.m. or 20:00 p.m., the correlation between the four meteorological elements and visibility is not significant (Figure 4). The Spearman correlation coefficients between airport visibility and temperature range from −0.359 at LXA to 0.704 at HAK and from −0.352 at LXA to 0.666 at URC at 8:00 a.m. and 20:00 p.m., respectively (Figure 4a,e). The visibility of most airports shows no significant positive correlation with the temperature within 0th–30th minutes. The greatest positive correlation between visibility and temperature within 0th–30th minutes occurred at HAK in Haikou and URC in Urumqi. While greater negative correlation within the 0th–30th minutes occurred at LXA in Lhasa, LJG in Lijiang and PEK in Beijing. The strong correlation between temperature and airport visibility is mainly concentrated in the 0th–30th minutes, and then the small correlation coefficient or the reversal of the positive and negative correlation in the 30th–60th minutes both show the weakening of the correlation. The Spearman correlation coefficients between airport visibility and air pressure range from −0.629 at HAK to 0.423 at URC and from −0.569 at HAK to 0.471 at URC at 8:00 a.m. and 20:00 p.m., respectively (Figure 4b,f). Within 5 min of vertical rise (within about 2000 gpm from the ground), the visibility has a strong correlation with the air pressure. Additionally, at most airports, the lower the air pressure under 2000 gpm from the ground, the better the visibility. After exceeding the range of 2000 gpm, the correlation between visibility and air pressure at most airports is weakened, but the positive and negative correlations at some airports are reversed, such as URC, HAK, and HRB in Harbin.

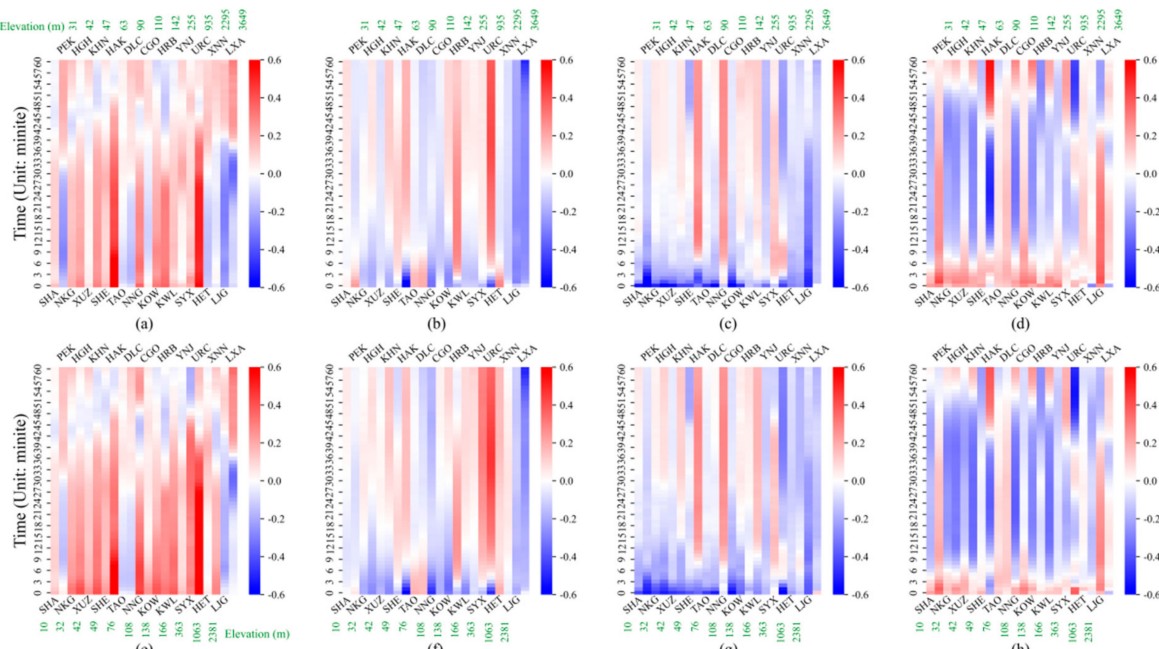

**Figure 4.** Spearman correlation coefficient between meteorological elements and airport visibility in every minute during 2010–2020. From left to right are the correlation coefficients of temperature (**a**,**e**), air pressure (**b**,**f**), relative humidity (**c**,**g**) and wind speed (**d**,**h**) with visibility. The first row and the second row represent 8:00 a.m. and 20:00 p.m., respectively. The airports are arranged left-to-right by altitude along the *x*-axis.

The Spearman correlation coefficients between airport visibility and relative humidity range from −0.672 at CGO to 0.322 at HAK and from −0.625 at URC to 0.289 at NNG at 8:00 a.m. and 20:00 p.m., respectively (Figure 4c,g). In almost all airports, the visibility is negatively correlated with the relative humidity within 400 gpm from the ground (within 2 min of vertical rise), that is, the greater the relative humidity, the poorer the visibility of the airports, and the negative correlation is greater in airports with lower altitude. With the increase in vertical height, the correlation between relative humidity and airport visibility

is weakened, and a few airports have positive and negative correlation reversal. However, since the detection equipment is far away from the airport due to horizontal movement during the ascent, the correlation of higher positions seems to be less referential. The Spearman correlation coefficients between airport visibility and wind speed range from −0.514 at HAK to 0.544 at HAK and from −0.540 at URC to 0.398 at HAK at 8:00 a.m. and 20:00 p.m., respectively (Figure 4d,h). At 8:00 a.m., the visibility is positively correlated with the wind speed within 2000 gpm from the ground (within 5 min of vertical rise) at most airports. After exceeding the range of 2000 gpm, the majority correlation coefficients are negative. Different from 8:00 a.m., at 20:00 p.m., the positive correlation mainly appear within 400 gpm from the ground (within 3 min of vertical rise) at most airports. HAK in Haikou, XNN in Xining and LXA in Lhasa are among the few airports where the visibility is negatively correlated with the wind speed within 300 gpm.

### 3.3. Hourly Airport Visibility Prediction by Artificial Intelligence Methods

### 3.3.1. Model Training

Based on meteorological data and 9 machine learning algorithms, the visibility prediction models of 47 airports are established independently. Hourly samples of 26 meteorological elements in 2018–2019 are used to train the visibility prediction models proposed in this study. On the whole, the hourly visibility prediction model based on various methods during the training period has good results, the Spearman correlation coefficients are mostly higher than 0.7, and the standard deviation ratios are around 1 (Figure 5). The standard deviation ratios of PAR- and IST-based models are not concentrated around 1, indicating that there is a certain gap between the simulations and the observations during the training period.

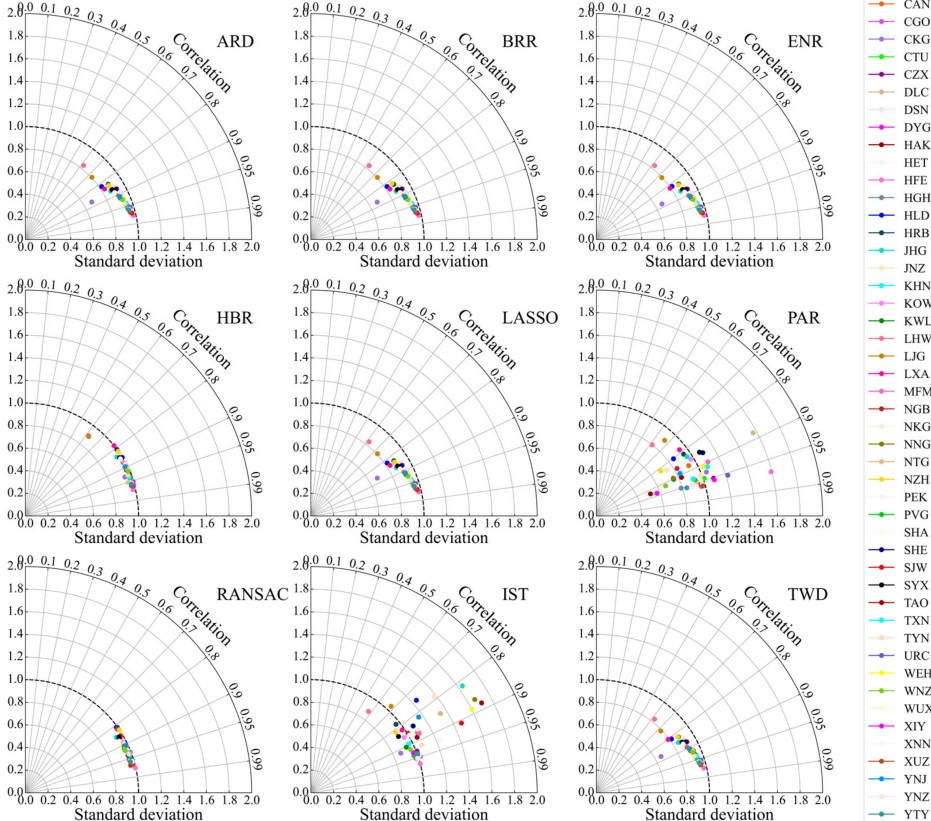

**Figure 5.** Taylor diagram presents a comparison of the hourly predicted and observed airports visibility in the training period (2018–2019). The diagram shows the correlation (the arc coordinate) and ratio of the standard deviation (the abscissa and ordinate) between the hourly prediction and observation.

From the RMSE (the root mean square error) and MAE (the mean absolute error) comparison between the simulations and observations in various months, it can also be seen that In addition to the poor results of PAR- and IST-based models, the other seven algorithm models perform well and have little difference in results (Figure 6). From June to September, the results of PAR- and IST-based models are worse than those in other months. The performances of TWD-, BRR-, ARD- and ENR-based models are relatively similar, and the distributions of RMSE and MAE of the four models are relatively similar. This shows that most models are well trained and suitable for hourly visibility prediction at all selected airports.

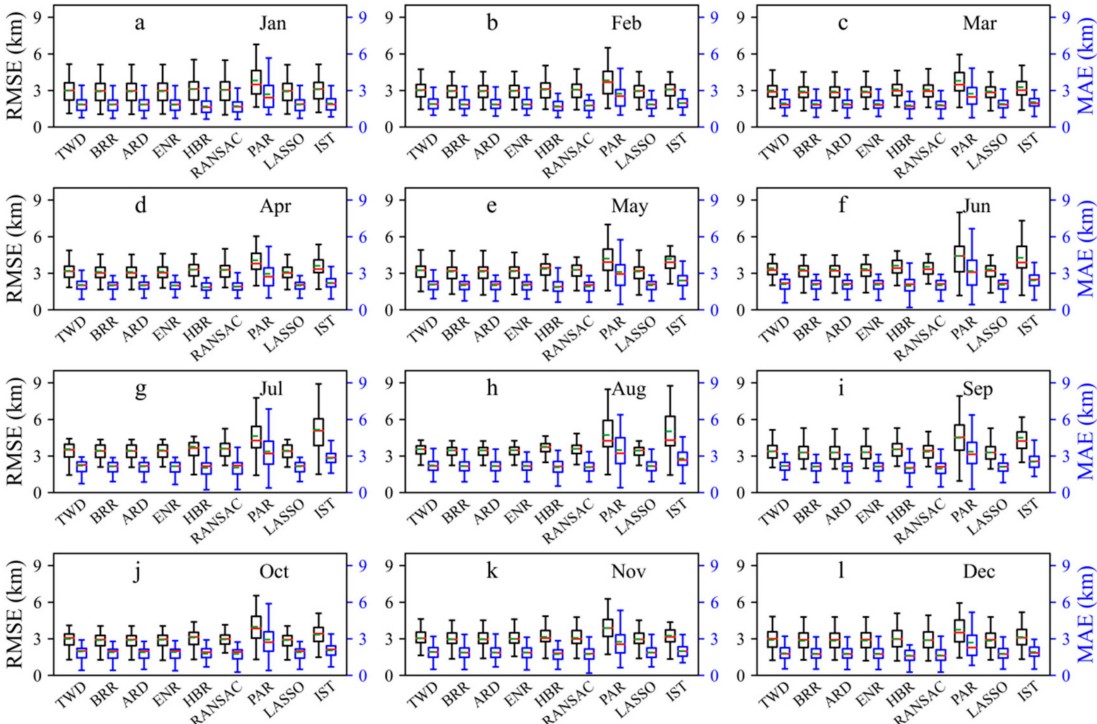

**Figure 6.** RMSE (black) and MAE (blue) between hourly visibility observations and predictions at 47 airports in the training period. (**a**–**l**) refers to January to December respectively. The red solid line and green dotted line represent the median and mean of each algorithm model at 47 airports, respectively. 25th and 75th percentiles are the bottom and top boundaries of a box, minimum and maximum are the bottom and top whiskers of a box. The notation applies for following box figures.

3.3.2. Model Testing

The results in testing period of each algorithm model are worse than the training period (Figures 7 and 8). Similar to the performance in the training period, there is a certain gap between the dispersion of the airport visibility prediction results obtained by PAR- and IST-based models and the observations, especially PAR-based models (Figure 7). Comparing the dispersion degree of the prediction and the observation, the dispersion degree of the visibility simulation results obtained by HBR- and RANSAC-based models is relatively consistent with the observations. The visibility prediction performance of ARD-, BRR-, ENR-, LASSO- and TWD-based models is relatively similar, and the distribution of 47 airports on the Taylor diagram of the corresponding algorithm is relatively consistent. At the same time, it can be seen that the airports with better visibility prediction results by different methods, while the airports with the worst prediction results are also fixed. This indicates that the geographical location of the airport may determine whether the visibility appears regularly, whether there are more complex inducements and other factors that affect the prediction results. For the airports located in Shijiazhuang (SJW), Hefei (HFE), Zhengzhou (CGO), Xuzhou (XUZ), Xianyang (XIY), Nanjing (NKG) and Hangzhou (HGH),

the visibility can be accurately predicted by the algorithm models, while for the airport located in Lhasa (LXA), Lanzhou (LHW), Manzhouli (NZH), Hohhot (HET), Shenyang (SHE), the visibility prediction performance are poor. From the visibility prediction results in different months, the RMSE and MAE between the predicted and observed are the smallest in November and December, while the they are the largest from June to September (Figure 8). RMSE and MAE between the prediction by PAR- and IST-based models and observation are large in all months.

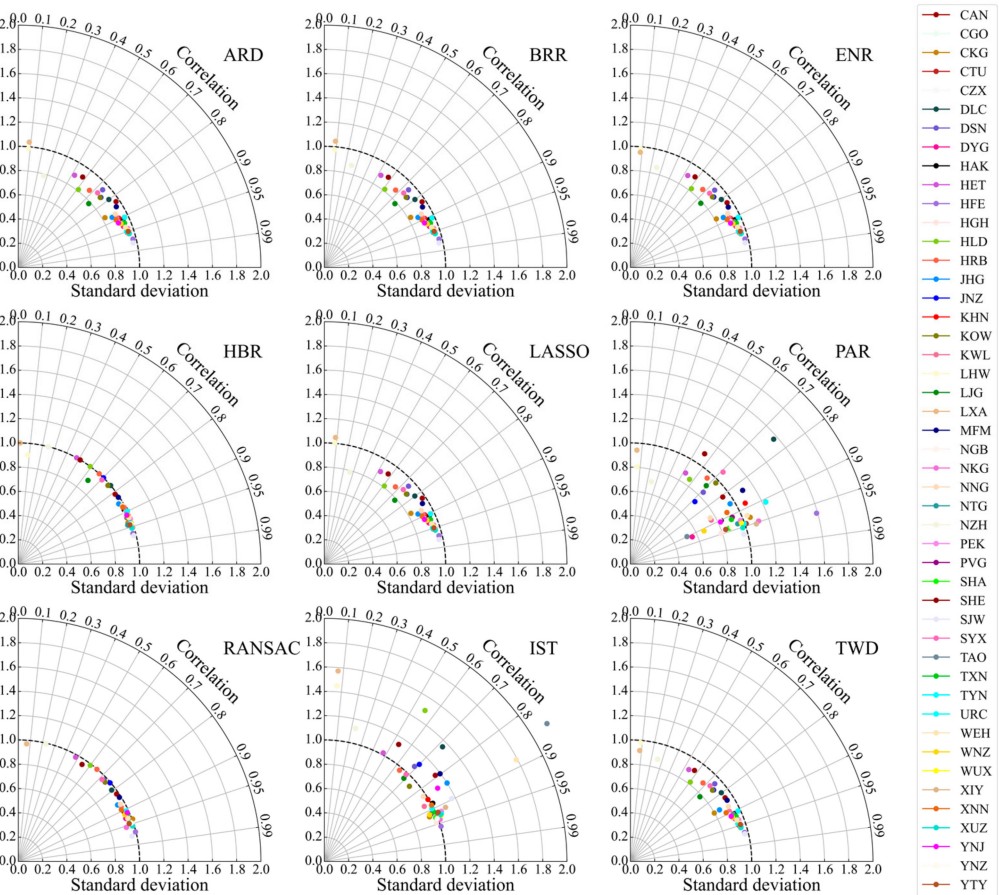

**Figure 7.** Taylor diagram presents a comparison of the hourly predicted and observed airports visibility in the testing period (2020). The diagram shows the correlation (the arc coordinate) and ratio of the standard deviation (the abscissa and ordinate) between the hourly prediction and observation.

Through kurtosis and skewness coefficients, the shape property of observation and prediction can be compared (Figures 9 and 10). In addition to the visibility observation of LHW, LXA, NZH, HET, HLD, SHE, HRB and CKG, the hourly visibility of other airports in 2020 shows a thin tail distribution (kurtosis coefficient > 3) in terms of kurtosis (Figure 9). For the eight airports with thick tailed observations (kurtosis coefficient < 3), the kurtosis coefficient of PAR- and HRB-based models prediction are close to the observed values, especially for the three airports, LHW, LXA and CKG with large observation kurtosis coefficients. For most airports, the kurtosis of visibility prediction obtained by all algorithm models is not much different from the kurtosis of observations. Among them, the kurtosis of the prediction results of all nine algorithm models at HEF, HGH, KWL, NGB, NTG, PVG, SHA, URC and WNZ are very close to the observations. Compare the prediction results of the models, IST-based model shows the worst effect. As for many airports, the kurtosis of the predicted results of IST-based model is greatly different from the kurtosis of the observation and the results of other algorithm models.

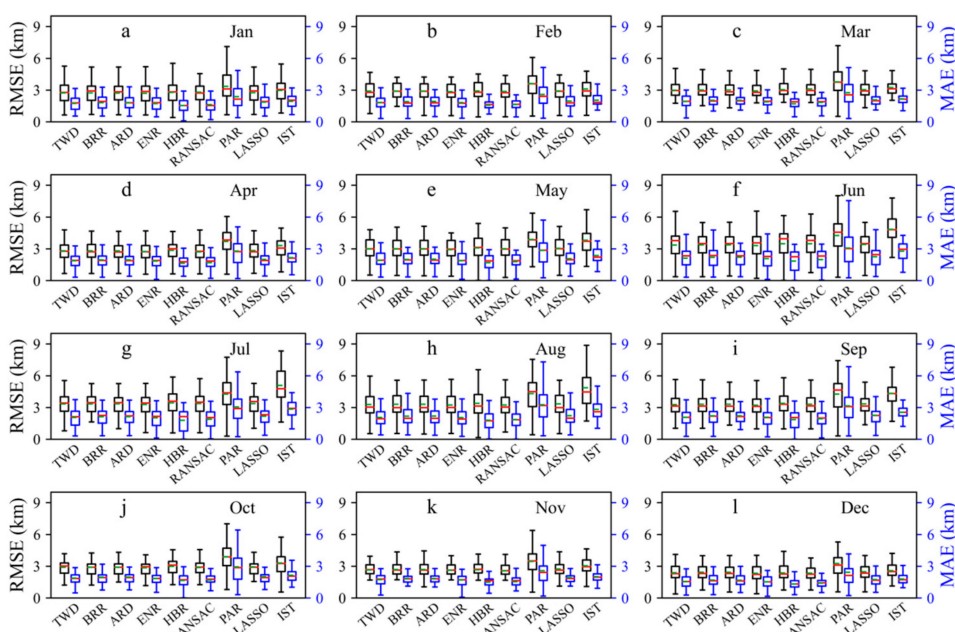

**Figure 8.** RMSE (black) and MAE (blue) between hourly visibility observations and predictions at 47 airports in the testing period. (**a**–**l**) refers to January to December respectively.

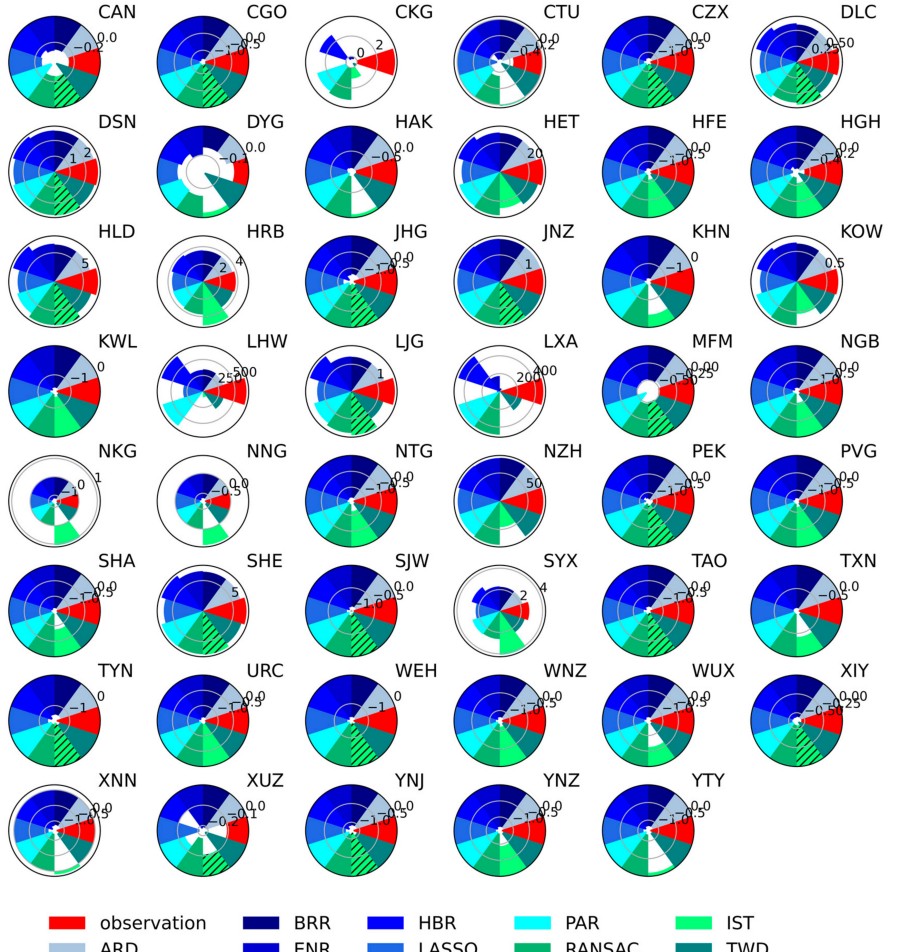

**Figure 9.** Kurtosis comparison between the hourly visibility observations and predictions at 47 airports in the testing period. The slash shadow represents that the result of this method is much worse than the observation and other methods.

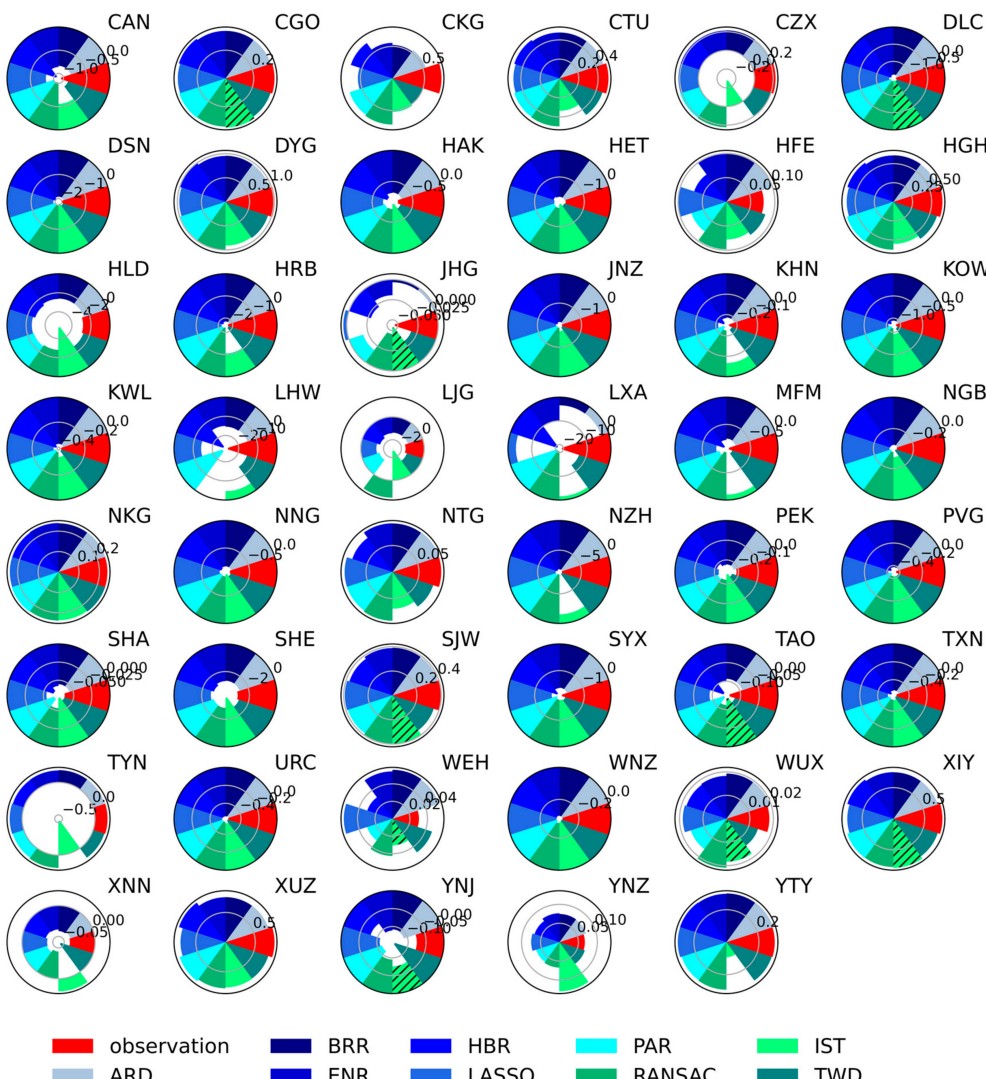

**Figure 10.** Skewness comparison between the hourly visibility observations and predictions at 47 airports in the testing period. The slash shadow represents that the result of this method is much worse than the observation and other methods.

Unlike the thin tailed distribution in airports visibility observations, which accounts for the vast majority of kurtosis, the ratio of the number of airports with negative (skewness coefficient < 0) and positive (skewness coefficient > 0) skew distribution is about 2:1 (Figure 10). For the hourly airport visibility observations in 2020, the maximum and minimum values of the skewness coefficient occur at DYG in Zhangjiajie and LHW in Lanzhou, respectively. There is little difference in the skewness coefficients of the airports with positive skew distribution of visibility, while in the negative skew distribution, the negative skewness coefficients of LHW, LXA and NZH are obviously higher than others. The skewness of the prediction results of all nine algorithm models at DSN, DYG, HGH, JNZ, NKG, URC, WNZ and XUZ are very close to the observations. The skewness coefficient of IST-based model results is better than kurtosis, though the skewness of IST-based model predictions is still the worst among all algorithm models.

## 4. Discussions

### 4.1. Airports with Typical Relationship between Visibility and Sounding Data

From the relationship between vertical meteorological data and airport visibility, HAK (in Haikou) is one of the special airports. The visibility of HAK has the greatest positive correlation with air temperature, relative humidity (within 400 gpm from the ground) and

wind speed (within 2000 gpm from the ground at 8:00 a.m. and 400 gpm from the ground at 20:00 p.m.) and the greatest negative correlation with air pressure (within about 2000 gpm from the ground) among all airports. HAK is located on Hainan Island in the tropical marine monsoon climate zone, and the climate is very complex, which should be the main reason for the typical visibility of HAK. Hainan Island is one of the regions with the highest rainfall in the world at the same latitude. However, the total precipitation is large but the spatial and temporal distribution is uneven. The rainy season generally occurs from May to October, and the dry season is from November to next April. The precipitation in rainy season accounts for about 80% of the annual. In winter, the cold air from the north to the south is blocked by Wuzhi Mountain, forming a static front near Haikou, thus causing low visibility events at HAK. At the altitude of 925 hPa, when the high-pressure system is in the north of HAK and the airport is controlled by the northeast wind, the visibility is better, and the visibility is poor when the airport is affected by the southeast wind and the humid and hot wind mass on the sea [33]. The high relative humidity in the atmosphere caused by the climate of Haikou is also an important factor affecting the visibility of HAK, because the increase in the relative humidity in the atmosphere will promote the hygroscopic growth of hygroscopic particles, increase the scattering cross section, and then lead to the reduction of visibility [34,35].

Besides HAK, LXA in Lhasa and URC in Urumqi are also two typical airports. LXA is among the few airports where the visibility is negatively correlated with the wind speed within 300 gpm. Which may be related to the fact that LXA is located in the Tibetan Plateau with special climate. The floating dust and sand rising with the increase in wind speed contribute to the poor visibility of LXA [17]. Cui et al. [36] point out that the biomass and incense burned by religious ceremonies during the day have made great contributions to the low visibility in Lhasa. Although it can try to explain the negative correlation between visibility and temperature at LXA, it seems that the religious burning of biomass and incense is mainly concentrated in densely populated urban and rural areas, which will not have a great impact on airport visibility, and it is difficult to explain the negative correlation between wind speed and airport visibility.

Compared with other airports, the visibility of URC is more strongly affected by the temperature. On the one hand, the comparison between winter and summer may be a factor reflecting the influence of the temperature on the visibility of URC. Urumqi is located at the rear of the Mongolian cold high in winter. When the stable snow is formed, the inversion layer is often maintained over the Junggar basin, resulting in the cloudy fog weather in the Junggar Basin and the northern Xinjiang along the Tianshan Mountains [37]. On the other hand, the temperature dew point difference directly determined by the temperature is also a factor that determines the visibility of URC. Liang [38] believes that the visibility is inversely related to the temperature and dew point difference at URC. The smaller difference, the easier it is to cause low visibility events.

*4.2. Comparison of Airport Visibility Prediction Models*

The prediction results of PAR- and IST-based models are worse than those of other models. For PAR-based model, the PAR reflects an online learning algorithm, that is, the data is input one by one, and the result is related to the arrangement data of the data. Therefore, the PAR-based model is unstable and cannot consider the whole situation. In addition, PAR-based model is an online learning algorithm for large-scale data. Its advantage is that it does not need all the data, and the model is updated by analyzing new samples. It has a fast training speed. However, for the meteorological prediction that rely heavily on large-scale historical data, the past periodic data patterns may be difficult to obtain effectively, and ultimately lead to poor prediction results. The above-mentioned defects may be the reason for the poor prediction results of the model. IST does not require the output characteristics of the model, and it is a nonparametric method for fitting monotonic models to data. However, this method is prone to computational difficulty and statistical overfitting problems in higher dimensions [39–41]. Moreover, IST looks for

a group of piecewise linear continuous functions that are not decreasing. However, for meteorological data, such as precipitation, wind speed and other characteristics change periodically with seasons, the data distribution assumed by the model is not consistent with the actual data distribution, which affects the prediction effect.

In addition to PAR- and IST-based models, the simulation results of other models are basically similar and the results are also good, which indicates that the other seven algorithm models can be better applied to airport visibility prediction. From the comparison of the dispersion of prediction and observation, the visibility prediction based on HBR and RANSAC are better. The advantage of good robustness of RANSAC algorithm may make the RANSAC-based model has better prediction performance, that is, the model parameters can be estimated even if there is a large degree of separation. The Huber loss function with strong tolerance to noise can better suppress the influence of outliers on the calculation results, thus making the HBR-based model results more accurate [42].

In addition to the different prediction performance caused by the model's own attributes, the input data used in this study are all meteorological elements, which may also be an important factor affecting the prediction results. Because human activities also make a great contribution to the change in visibility. In the Section 4.1, it is mentioned that the collective burning of incense in religious areas may lead to a decline in visibility [36]. Moreover, such human activity factors with obvious time characteristics should be taken into account in the input data of the model, such as the changes in visibility caused by the massive burning of fossil fuels for heating in northern China in cold seasons [43]. Industry, energy consumption, vehicles and other socioeconomic factors are significantly associated with atmospheric aerosols [44–47], and the increase in aerosols will directly lead to the decrease of visibility. Therefore, in the future model construction, the environmental factors that affect visibility and changed by human activities mentioned in the previous study, such as PM2.5, PM10, $NO_2$, $SO_2$, $O_3$ and CO, should be considered at the same time to improve the prediction accuracy of the model.

## 5. Conclusions

In this study, L-band sounding data during 2010–2020 is employed to investigate vertical characteristics of temperature, air pressure, and relative humidity wind speed at the selected airports. The relationship between airport visibility and meteorological elements at different potential heights are also examined. Then, based on the hourly 26 meteorological elements in 2018–2020, the hourly visibility of the airports are predicted by 9 artificial intelligence algorithm models, and the prediction results of different methods are compared. The major findings are summarized as follows.

(1) For the vertical change in airport meteorological elements, the negative vertical trends of temperature and relative humidity have an obvious pattern of getting greater from northwestern to southeastern China. In eastern and western China, the vertical trend ranges of air pressure are different, although both are significantly decreased, and the negative trend in the eastern is greater.

(2) Within about 2000 gpm from the ground, the visibility has a strong correlation with the air pressure and most of them are negative. The visibility is negatively correlated with the relative humidity within 400 gpm from the ground. At 8:00 a.m., the visibility is positively correlated with the wind speed within 2000 gpm from the ground at most airports, while at 20:00 p.m., the positive correlation mainly appear within 400 gpm from the ground.

(3) There is a certain gap between the dispersion of the airport visibility prediction results obtained with the PAR- and IST-based models and the observations. Comparing the dispersion degree of the prediction and the observation, the dispersion degree of the visibility simulation results obtained by HBR- and RANSAC-based models is relatively consistent with the observations. For the airports located in Shijiazhuang (SJW), Hefei (HFE), Zhengzhou (CGO), Xuzhou (XUZ), Xianyang (XIY), Nanjing (NKG), and Hangzhou (HGH), the visibility can be accurately predicted by the algorithm models,

while for the airport located in Lhasa (LXA), Lanzhou (LHW), Manzhouli (NZH), Hohhot (HET), and Shenyang (SHE), the visibility prediction performance are poor.

Both meteorological element prediction and climate prediction are extremely complicated. The improvement of forecast accuracy depends on modern observation and advanced methods. In this context, this study conducts airport visibility forecast based on the current relatively foreword AI method and the current relatively complete and comprehensive meteorological observation. In order to carry out targeted research on 47 airports, the visibility prediction model most suitable for each airport is found through the performance of nine different algorithms in different airports. This study can provide support for the climate prediction around the airport in the following aspects: (1) By establishing a visibility prediction system, the climate over selected airport in 2018–2020 has been combed and analyzed in a targeted way, and the meteorological change patterns of airports in different climate zones have been strengthened. (2) Although machine learning methods are widely used at present, they still have limitations. Based on this study, subsequent studies can add interpretable factors such as aerodynamics to improve the ability of prediction and early warning.

**Author Contributions:** Writing—original draft, J.D.; Writing—review & editing, J.D; Supervision, G.Z. and J.Y.; Data curation, S.W., B.X., R.J., J.G. and K.W.; Methodology, X.D.; Visualization, J.D. and Y.T. All authors have read and agreed to the published version of the manuscript.

**Funding:** This research was funded by by the National Key Research and Development program of China (2020YFB1600103) and the National Natural Science Foundation of China (grants 41871020).

**Institutional Review Board Statement:** Not applicable.

**Informed Consent Statement:** Not applicable.

**Data Availability Statement:** The study did not report any data.

**Acknowledgments:** Thank each of the authors for their contributions to this study. At the same time, we would like to thank each editor and reviewer for their constructive comments and revisions on the publication of this article.

**Conflicts of Interest:** The authors declare that they have no conflict of interest.

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
