# Peer review of "Temporal and Spatial Characteristics of Meteorological Elements in the Vertical Direction at Airports and Hourly Airport Visibility Prediction by Artificial Intelligence Methods"

_sustainability, doi:10.3390/su141912213_

Round 1

Reviewer 1 Report

Review of: Temporal and spatial characteristics of meteorological elements in vertical direction at airports and hourly airport visibility prediction by artificial intelligence methods

By: Jin Ding, Guoping Zhang, Jing Yang, Shudong Wang, Bing Xue, Xiangyu Du, Ye Tian, Kuoyin Wang, Ruijiao Jiang and Jinbing Gao

Overview

This study analyzed the L-band sounding data collected at airports in 2010-2020 and investigated the relationship between airport visibility and meteorological elements at different potential heights. The paper is well written and the analysis is comprehensive and thorough. I recommend minor revisions after addressing the specific comments and questions below.

Specific comments:

1.     The present study focused on the relationship between visibility and the meteorological elements, while the visibility can also be affected by the emissions from human activities, which is suggested to be discussed in the manuscript.

2.     The abbreviations like “PAR” and “IST” in the abstract and “LSTM model” (L74) should be defined at first use.

3.     L305: One of the word “is” should be deleted.

4.     L459-460, L487: Both sentences are difficult to be understood, which are suggested to be rewritten.

Author Response

Dear Editors and Reviewers,

Thank you for your comments and suggestions concerning our manuscript entitled ‘Temporal and spatial characteristics of meteorological elements in vertical direction at airports and hourly airport visibility prediction by artificial intelligence methods’ (Submission ID: sustainability-1896972). Those comments have been very helpful for us to revise and improve our manuscript. We have carefully considered these comments and suggestions, and made revisions accordingly. We hope that the revised manuscript has satisfactorily addressed your concerns and that it meets the requirements for publication in Sustainability. In the following, we provide our point-to-point responses to Reviewers’ comments.

Reviewer #1:

Overall comments: This study analyzed the L-band sounding data collected at airports in 2010-2020 and investigated the relationship between airport visibility and meteorological elements at different potential heights. The paper is well written and the analysis is comprehensive and thorough. I recommend minor revisions after addressing the specific comments and questions below.

Comment 1: The present study focused on the relationship between visibility and the meteorological elements, while the visibility can also be affected by the emissions from human activities, which is suggested to be discussed in the manuscript.

Response 1: Thanks the reviewer for this comment. After reading relevant literature, according to the comment of reviewers, we added the impact of human activities on visibility in the discussion section, as well as the environmental factors affecting visibility caused by human activities in the future modeling process. Please check L606-619. The added literature are as follows:

Wang, K. , Wang, W. , Li, L. , Li, J. , & Jiang, J. . (2020). Seasonal concentration distribution of pm1.0 and pm2.5 and a risk assessment of bound trace metals in harbin, china: effect of the species distribution of heavy metals and heat supply. Scientific Reports, 10(1).  https://doi.org/10.1038/s41598-020-65187-7

Chen, X. , Li, X. , Yuan, X. , Zeng, G. , & Chen, G. . (2018). Effects of human activities and climate change on the reduction of visibility in beijing over the past 36 years. Environment international, 116, 92-100. https://doi.org/10.1016/j.envint.2018.04.009

Wiston, M. . (2017). Status of air pollution in botswana and significance to air quality and human health. Journal of Health and Pollution, 8(15), 15-24. https://doi.org/10.5696/2156-9614-8.15.15

Irani, T. , Amiri, H. , & Deyhim, H. . (2021). Evaluating visibility range on air pollution using narx neural network. https://doi.org/10.47277/JETT/9(2)547

Lee, H. H. , Iraqui, O. , Gu, Y. , Yim, H. , & Wang, C. . (2017). Impacts of air pollutants from fire and non-fire emissions on the regional air quality in southeast asia. Atmospheric Chemistry & Physics, 1-53. https://doi.org/10.5194/acp-2017-1096

Comment 2: The abbreviations like “PAR” and “IST” in the abstract and “LSTM model” (L74) should be defined at first use.

Response 2: We are very grateful to the reviewers for specifying the full name of each abbreviation in the abstract and main text for the first time. The full names of abbreviations appearing for the first time in the abstract and main text have been added. Please check the abstract and L77.

Comment 3: L305: One of the word “is” should be deleted.

Response 3: We have accepted the suggestion and remove the word ‘is’ and re-edited the sentence. Please check L305-311.

Comment 4: L459-460, L487: Both sentences are difficult to be understood, which are suggested to be rewritten..

Response 4: Thanks the reviewer for this comment. For the above sentence, we reorganized the sentence to make the content brief and accurate. Please check L463-466 and L491-493.

Reviewer 2 Report

Although the manuscript has the potential to advance our understanding of temporal and spatial characteristics of meteorological elements, the explanation and figure require extensive revision. Please address the comments to improve the quality of your article.

1. Why would somebody employ " 9 artificial intelligence algorithm models" ? Are you able to address the applicability of each of these in a table? Which one is better? I don't see any comparable figures!

2. I believe section 2.2.3 " kurtosis and skewness coefficient" is unnecessary! This is an established fact. Please make it short. 

3. Please include some recommendations regarding how this study could be utilized to improve climate forecasting near the airport. 

Author Response

Dear Editors and Reviewers,

Thank you for your comments and suggestions concerning our manuscript entitled ‘Temporal and spatial characteristics of meteorological elements in vertical direction at airports and hourly airport visibility prediction by artificial intelligence methods’ (Submission ID: sustainability-1896972). Those comments have been very helpful for us to revise and improve our manuscript. We have carefully considered these comments and suggestions, and made revisions accordingly. We hope that the revised manuscript has satisfactorily addressed your concerns and that it meets the requirements for publication in Sustainability. In the following, we provide our point-to-point responses to Reviewers’ comments.

Reviewer #2:

Overall comments: Although the manuscript has the potential to advance our understanding of temporal and spatial characteristics of meteorological elements, the explanation and figure require extensive revision. Please address the comments to improve the quality of your article.

Comment 1: Why would somebody employ " 9 artificial intelligence algorithm models" ? Are you able to address the applicability of each of these in a table? Which one is better? I don't see any comparable figures!

Response 1: Thanks the reviewer for this comment. Because the selected airports are distributed in different climate zones, and their climate characteristics, geographical characteristics, population density and economic conditions are different from each other, if only one method is selected for visibility prediction model modeling, on the one hand, the model most suitable for each airport cannot be found, and on the other hand, the optimal prediction results may not be obtained due to the limitations of the model attributes. 

Also we have accepted the suggestion to address the applicability of artificial intelligence algorithm models in a table. Table 3 lists the nine algorithms used in the study and describes their applicability, so as to increase the comparison between methods and show more details of algorithms. Please check Table 3.

As for the performance comparison of several models, Figs. 5-10 illustrate MAE, RMSE, correlation coefficient and ratio of the standard deviation respectively. The comparison results are also stated in the Results section, and then the Discussion section also tries to explain why different algorithm models behave differently. Therefore, we believe that the performance comparison description of different algorithm model simulation results in our paper is relatively complete and clear, and it is not necessary to use charts to further explain.

Comment 2: I believe section 2.2.3 " kurtosis and skewness coefficient" is unnecessary! This is an established fact. Please make it short.

Response 2: Thanks the reviewer for this comment. As we said in the paper, kurtosis and skewness can reflect the shape characteristics of observations and predictions. It is indeed an established fact for the observations, but for the predicted values obtained from different models, the possible kurtosis and skewness coefficients are obviously different from the observations, especially the series composed of predicted values that increase with time. With the increase of prediction time, longer and longer prediction result series will show their own shape characteristics. If the difference of kurtosis coefficient and skewness coefficient between the predicted series and the observed series is small, it indicates that the prediction model has good stability and accuracy, and vice versa. To sum up, we think it is necessary to compare the kurtosis and skewness of observation data and prediction data.

Comment 3: Please include some recommendations regarding how this study could be utilized to improve climate forecasting near the airport.

Response 3: Thanks the reviewer for this comment. This study not only contributes to airport visibility prediction, but also helps to predict the climate around the airport. For example, 1) by establishing a visibility forecast system, the climate of 2018-2020 has been combed and analyzed in a targeted way, and the meteorological change rules of airports in different climate zones have been strengthened. 2) Although machine learning methods are widely used at present, they still have limitations. Based on this study, subsequent studies can add interpretable factors such as aerodynamics to improve the ability of prediction and early warning. We have added a more detailed description in the Conclusion section, please check L649-662.

Round 2

Reviewer 2 Report

Thanks for the revision.